# Micro-Vision Based High-Precision Space Assembly Approach for Trans-Scale Micro-Device: The CFTA Example

**DOI:** 10.3390/s23010450

**Published:** 2023-01-01

**Authors:** Juan Zhang, Xi Dai, Wenrong Wu, Kai Du

**Affiliations:** Research Center of Laser Fusion, China Academy of Engineering Physics, Mianyang 621900, China

**Keywords:** micro-vision, multi-vision monitoring model, posture alignment, glue mass control, trans-scale micro-device assembly, micro-assembly

## Abstract

For assembly of trans-scale micro-device capsule fill tube assemblies (CFTA) for inertial confinement fusion (ICF) targets, a high-precision space assembly approach based on micro-vision is proposed in this paper. The approach consists of three modules: (i) a posture alignment module based on a multi-vision monitoring model that is designed to align two trans-scale micro-parts in 5DOF while one micro-part is in ten microns and the other one is in hundreds of microns; (ii) an insertion depth control module based on a proposed local deformation detection method to control micro-part insertion depth; (iii) a glue mass control module based on simulation research that is designed to control glue mass quantitatively and to bond micro-parts together. A series of experiments were conducted and experimental results reveal that attitude alignment control error is less than ±0.3°, position alignment control error is less than ±5 µm, and insertion depth control error is less than ±5 μm. Deviation of glue spot diameter is controlled at less than 15 μm. A CFTA was assembled based on the proposed approach, the position error in 3D space measured by computerized tomography (CT) is less than 5 μm, and glue spot diameter at the joint is 56 μm. Through volume calculation by the cone calculation formula, the glue mass is about 23 PL when the cone height is half the diameter.

## 1. Introduction

Micro-assembly is used to manipulate millimeter-sized or smaller components with micron accuracy. Recently, the micro-assembly technique has been widely applied to fields including aerospace, biomedical, materials science, and others MEMS, and micro-assembly robots have been developed for assembling micro-parts [1,2,3]. Micro-assembly robots are usually equipped with micro-vision for monitoring the assembly process [4,5,6]. However, micro-vision is problematic in that the larger the magnification, the smaller the field of view (FOV), that is, there is a contradiction between high precision detection and wide field of view. For trans-scale micro-device assembly, larger-scale parts are usually not fully observed and parts easily move out of the FOV during the posture adjustment process.

For trans-scale micro-device assembly, a series of studies have been carried out. Typically, through two-dimensional translation motion during the micro-part attitude adjustment process, micro-part can be always kept in the FOV during the assembly process [7]. Later, an active zoom strategy to resolve any contradiction between FOV and magnification was proposed [8]. The strategy includes coarse and fine adjustment processes. In the coarse adjustment process, zoom speed is controlled by the image position and focus function value of the micro-part. In the fine adjustment process, parts are assembled in high magnification. Furthermore, a zoom control model based on artificial potential field was built to realize automatic zoom of micro-vision during the hole and pole assembly process [9]. Recently, a four-way micro-vision system for trans-scale micro-device assembly was established and a trans-scale assembly method for micro-sphere and micro-particle was proposed [10].

As listed above, the trans-scale assembly research mainly focused on three degrees of freedom (DOF) posture detection and alignment control; six DOF (6DOF) posture control is less researched. In addition, the insertion depth control and micro-parts bonding control, which affect assembly accuracy during the assembly process, are rarely mentioned.

ICF is an effective way to achieve nuclear fusion and ICF research requires different types of targets. For target fabrication, a micro-sphere is commonly used as the fuel container, and a micro-tube is used as the transport channel for fuel. CFTA is a trans-scale micro-device composed of a micro-sphere and a micro-tube and is an important composer for the ICF target. The micro-sphere scale is usually in hundreds of microns, while the micro-tube scale is in ten microns. The micro-sphere is composed of a glow discharge hydrocarbon polymer (GDP) material, and the micro-tube is a transparent glass material.

Due to the contradiction between visual high precision and wide field of view, firstly, it is difficult to detect the 6DOF posture of a trans-scale micro-sphere and micro-tube. Secondly, the micro-tube occludes the micro-pore, which is at the top of the micro-sphere in the insertion process. As the micro-tube is transparent, that image information superposition occurs, and the micro-tube end position is hard to detect. As a result, the insertion depth cannot be controlled with high precision. Thirdly, the glue mass at the joint cannot be controlled quantitatively, as the parameters for glue mass control are not clear.

To address these problems, in this paper, a high-precision assembly approach for CFTA includes posture alignment in 5DOF, insertion depth control, and quantitative bonding control of trans-scale micro-parts was studied. First, a posture alignment method based on a multi-vision monitoring model and detection method was proposed to detect and align the micro-parts in 5DOF. Second, an insertion depth control method based on local deformation detection was proposed to control insertion depth in the visual occlusion scenario. Third, the roles of different glue parameters for glue mass control were analyzed based on the simulation and the ability of the glue mass diameter to be controlled quantitatively.

The remainder of the paper is organized as follows. Section 2 introduces the CFTA assembly task, in detail, as well as the structure of the micro-assembly robot. Section 3 presents the trans-scale assembly method, including the posture alignment module, depth control module, and glue mass control module. Section 4 shows the experiment results. Finally, conclusions are provided in Section 5.

## 2. Task Specification and Micro-Assembly Robot System Design

### 2.1. Assembly Task Specification

The assembly task is to insert a micro-tube into a micro-sphere, and bond the two micro-parts together with glue. A diagram of the assembly process is shown in Figure 1. Firstly, the attitude of the micro-tube is adjusted to align with the micro-sphere, as shown in Figure 1a. Secondly, the position of the micro-tube is controlled to align with the micro-pore, which is at the top of the micro-sphere, as shown in Figure 1b. Thirdly, the micro-tube is inserted into the micro-sphere with control, as shown in Figure 1c. Lastly, glue is dispensed at the joint to bond the two parts together, as shown in Figure 1d.

According to different ICF needs, the outer diameter of the micro-sphere differs. It is usually in the hundreds of microns, and the diameter of the micro-pore, which is at the top of the micro-sphere, is usually in the tens of microns. The outer diameter of the micro-tube end which is inserted into the micro-pore is smaller than the diameter of micro-pore by about 2 μm~3 μm. The glue mass at the joint needs to be quantitatively controlled, usually at the level of tens of PL. The difficulties in the assembly process are listed below.

(1) Posture alignment. As there is a large size difference among micro-parts, it is not easy to assembly micro-parts with high precision due to the contradiction between high precision and wide field of view [11,12]. It is necessary to establish a new vision monitoring model to guide assembly, and a posture alignment method of trans-scale micro-parts should be studied;

(2) Insertion depth control. Because visual occlusion occurs during the micro-tube insertion process, it is difficult to image the micro-tube end; therefore, the insertion depth cannot be effectively detected and controlled. An insertion depth control method based on local deformation detection in the visual occlusion scenario was studied and is discussed in this paper;

(3) Glue mass control. As glue mass at the joint needs to be controlled quantitatively and its volume needs to be controlled at the tens of PL level, the time-pressure dispensing mode was selected due to its controllability. Parameters such as pressure, contact time of the dispensing needle, dispensing needle diameter, etc., concurrently influence the glue mass. However, the role of these different parameters is not clear, and quantitative control of the glue mass cannot be achieved. Thus, the role of different parameters needs to be analyzed based on a simulation for glue mass control.

### 2.2. Micro-Assembly Robot System Design

As there is a large size difference among micro-parts, it is difficult to observe them at the same magnification. Additionally, micro-pores are not easily observed from a horizontal perspective. For these problems, a micro-assembly robot system with multi-vision system was designed, as shown in Figure 2a. It mainly includes four parts: a multi-vision system, a micro-operating system, a dispensing system, and a software control system.

The multi-vision system consists of four microscopic vision systems, which are labeled as No. 1, No. 2, No. 3, and No. 4, respectively. No. 1 and No. 2 are arranged parallel to the horizontal plane, and the angle between their optical axes is 90 degrees. They both have lower magnification to obtain global information about the micro-parts. No. 3 is located on the top of No. 1, and the angle between its optical axis and the horizontal plane is 60 degrees. Similarly, No. 4 is located on the top of No. 2, and the angle between its optical axis and the horizontal plane is also 60 degrees. No. 3 and No. 4 both have higher magnification to obtain magnified local information about the micro-parts. The four microscopic vision all have three transitional DOF along X-, Y-, and Z-axes, which are used to adjust position to present micro-parts in FOV and focus. No. 3 and No. 4 additionally have one transitional DOF along the optical axis for more convenient focus.

The micro-operating system consists of a micro-manipulator and a console. The end of micro-manipulator connects to a micro-gripper to clamp the micro-tube, and the surface of the console connects to a micro-gripper to clamp the micro-sphere. The micro-manipulator and console both have three transitional DOF along the X-, Y-, and Z-axes, and two rotational DOF around the X- and Y-axes. The DOF of the micro-manipulator are used to adjust the posture of the micro-tube during the assembly process. The DOF of the console are used to adjust the initial posture of the micro-sphere and move the micro-sphere into the FOV of No. 1 and No. 2.

The dispensing system is used to control dispensing needle movement to the joint for dispensing. It has three translation DOF along the X-, Y-, and Z-axes, and two rotational DOF around the X- and Y-axes. A time–pressure dispenser is connected with a dispensing needle for glue mass control.

The software control system mainly includes an image acquisition and processing module, a movement control module, a micro-force acquisition module, and a workflow control module. The operation interface of software control system is shown in Figure 2b. The image acquisition and processing module includes CCD control, zoom lens control, and image display inference. The movement control module includes motion control of the multi-vision system, a micro-manipulator, a console, and a dispensing needle. The micro-force acquisition module collects force information in assembly process in real time. The workflow control module controls the assembly process in a semi-automatic mode.

## 3. High-Precision Assembly Approach Based on Collaboration of Posture Alignment, Insertion Depth Control and Glue Mass Control

The characteristics of the proposed high-precision assembly approach lies in the following aspects: (i) a multi-vision monitoring model based on the multi-vision system is built; (ii) posture alignment of trans-scale micro-parts based on the multi-vision monitoring model and posture detection method is realized; (iii) insertion depth is controlled based on the proposed local deformation detection method, abbreviated as LDD; (iv) a glue mass control strategy is proposed based on simulation research to control glue mass diameter quantitatively. The detailed process of this approach is described below.

### 3.1. Multi-Vision Monitoring Model

A sketch map of the monitoring model is shown in Figure 3. It is composed of a monitoring model of the horizontal microscopic vision and a monitoring model of the 60-degree microscopic vision.

The monitoring model of the horizontal microscopic vision is shown in Figure 3a. Planes *F*_1_ and *F*_2_ represent the focus planes of No. 1 and No. 2, respectively. It can be seen that No. 1 can detect the position of deviation of the micro-parts in the X-axis and Z-axis directions, and the attitude of deviation around the Y-axis. No. 2 can detect position deviation in the Y-axis and Z-axis directions, and the attitude of deviation around the X-axis.

The monitoring model of 60-degree microscopic vision is shown in Figure 3b. Plane *F*_3_ is the focus plane of No. 3. By controlling movement of No. 3 along its optical axis, the area (*a*) of the micro-pore can be monitored. Plane *F*_4_ is the focus plane of No. 4. By controlling the movement of No. 4 along its optical axis, the region (*b*) of the micro-pore can be monitored. Through detection results from No. 3 and No. 4, the center point of the micro-pore in Cartesian space can be calculated. Furthermore, No. 3 can detect the position of the micro-tube end in the X-axis and Z-axis, and No. 4 can detect the position of the micro-tube end in the Y-axis and Z-axis. Thus, the micro-tube movement can be controlled to the center of the micro-pore under the guidance of No. 3 and No. 4 with high precision.

### 3.2. Posture Alignment Module Based on Multi-Vision Monitoring Model

Based on the multi-vision monitoring model, the attitude of the micro-parts is finely aligned under the guidance of No. 1 and No. 2, while position is coarsely aligned. Then, position is finely aligned under the guidance of No. 3 and No. 4.

#### 3.2.1. Attitude Fine Alignment

The attitude increment of the micro-tube in Cartesian space is defined as $\dot{\delta }(t)$
*=* [Δ*δ_x_*(*t*), Δ*δ_y_*(*t*)]^T^, and the attitude increment in image space is defined as $\dot{\theta }(t)$
*=* [Δ*θ_x_*(*t*), Δ*θ_y_*(*t*)]^T^. The attitude increment from Cartesian space to image space can be calculated by Formula (1)
(1)$$\dot{\theta }(t)=J_{R}(\delta )\dot{\delta }(t)$$
where Δ*θ_x_*(*t*) and Δ*θ_y_*(*t*) represent the attitude increment around the X- and Y-axes in image space, respectively. Δ*δ_x_*(*t*) and Δ*δ_y_*(*t*) represent the attitude increment in Cartesian space. *J_R_* is the Jacobian matrix from Cartesian space to image space.

Through controlling the micro-tube to move *n* times around the X-axis and Y-axis in Cartesian space, *J_R_* can be calculated by fitting the method of least squares [13], as shown in Formula (2)
(2)$$\left[\begin{array}{cccc} \Delta \theta_{x}{}_{1} & \Delta \theta_{x}{}_{2} & \ldots & \Delta \theta_{x}{}_{n}\\ \Delta \theta_{y}{}_{1} & \Delta \theta_{y}{}_{2} & \ldots & \Delta \theta_{y}{}_{n} \end{array}\right]=J_{R}\left[\begin{array}{cccc} \Delta \delta_{x1} & \Delta \delta_{x2} & \ldots & \Delta \delta_{xn}\\ \Delta \delta_{y1} & \Delta \delta_{y2} & \ldots & \Delta \delta_{yn} \end{array}\right]$$
where Δ*δ_xi_* and Δ*δ_yi_* represent the *i*-th angle increment around X-axis and Y-axis in Cartesian space (*i* = 1, 2,..., *n*), respectively. Δ*θ_xi_* and Δ*θ_yi_* represent the *i*-th angle increment in the No.1 and No.2 image spaces, respectively, which are calculated by image feature extraction of the micro-tube end and the micro-pore.

Figure 4a shows the extracted image feature to calculate attitude increment. By extracting the central axis (*l_tm_*) of the micro-tube end and the central axis (*l_sm_*) of the micro-pore, the attitude error in the image space can be calculated. During the assembly process, the attitude increment in Cartesian space is calculated using Formula (3).
(3)$${\dot{\delta }}_{t}={\left(J_{R}^{T}J_{R}\right)}^{-1}J_{R}^{T}{\dot{\theta }}_{t}$$

Micro-tube attitude is adjusted by the P controller. The control law is shown in Formula (4), where Δ*u_a_*(*k*) *=* [Δ*u_ax_*(*k*), Δ*u_ay_*(*k*)]^T^ is the control value at *k*-*th* control circle. *K_Pa_* is the P coefficient. Δ*δ*(*k*) *=* [Δ*δ_x_*(*k*), Δ*δ_y_*(*k*)]^T^ is attitude error calculated using Formula (3). *e_a_*(*k*) *=* [Δ*α*(*k*), Δ*β*(*k*)]^T^ is attitude error calculated in image space using the image feature extraction method at *k*-*th* control circle. *e_Ta_ =* [*e_Tax_*, *e_Tay_*]^T^ is the set control threshold.
(4)$$\Delta u_{a}(k)=\left\{ \begin{array}{ll} K_{pa}\Delta \delta (k) & |e_{a}(k)|\geq e_{Ta}\\ 0 & |e_{a}(k)|\;<e_{Ta} \end{array}\right.$$

#### 3.2.2. Position Fine Alignment

The motion increment of the micro-tube in Cartesian space is defined as $\dot{p}(t)$ = [Δ*x*(*t*), Δ*y*(*t*), Δ*z*(*t*)]^T^, and the motion increment of the micro-tube in image space is defined as $\dot{s}(t)$ = [Δ*u*_1_(*t*), Δ*v*_1_(*t*), Δ*u*_2_(*t*), Δ*v*_2_(*t*)]^T^. (Δ*u*_1_(*t*), Δ*v*_1_(*t*)) and (Δ*u*_2_(*t*), Δ*v*_2_(*t*)) represent position increments in No. 3 and No. 4 image space, respectively. They are calculated by image feature extraction of the micro-tube end and the micro-pore.

Figure 4b shows the extracted image feature to calculate position increment. By extracting central point (*p_tm_*) of the micro-tube end and the central point (*p_sm_*) of the micro-pore, position increment in image space can be calculated. The relationship between the motion increment from Cartesian space and image space can be calculated using Formula (5).
(5)$$\dot{s}(t)=J_{G}(p)\dot{p}(t)$$
where *J_G_* represents the Jacobian matrix from Cartesian space to image space. Through controlling the micro-tube to move *m* times in Cartesian space, *J_G_* can be calculated by fitting the method of least squares [14,15], as shown in Formula (6).
(6)$$\left[\begin{array}{l} \Delta u_{11}\Delta u_{12}\ldots \Delta u_{1m}\\ \Delta v_{11}\Delta v_{12}\ldots \Delta v_{1m}\\ \Delta u_{21}\Delta u_{22}\ldots \Delta u_{2m}\\ \Delta v_{21}\Delta v_{22}\ldots \Delta v_{2m}\; \end{array}\right]=J_{G}\left[\begin{array}{cccc} \Delta x_{1} & \Delta x_{2} & \ldots & \Delta x_{m}\\ \Delta y_{1} & \Delta y_{2} & \ldots & \Delta y_{m}\\ \Delta z_{1} & \Delta z_{2} & \ldots & \Delta z_{m} \end{array}\right]$$
where (Δ*x_j_*, Δ*y_j_*, Δ*z_j_*) represents the *j*-*th* position increments of the micro-tube in Cartesian space (*j* = 1,2,...,*m*). (Δ*u*_1_*_j_*, Δ*v*_1_*_j_*) and (Δ*u*_2*j*_, Δ*v*_2_*_j_*) represent the *j*-th position increments in No. 1 and No. 2 image spaces, respectively. During the assembly process, position error in Cartesian space is calculated on-line using Formula (7).
(7)$$\dot{p}(t)={\left(J_{G}{}^{T}J_{G}\right)}^{-1}J_{G}{}^{T}\dot{s}(t)$$

Micro-tube position is also adjusted by the P controller. The control law is shown in Formula (8), where Δ*u_p_*(*k*) *=* [Δ*u_px_*(*k*), Δ*u_py_*(*k*), Δ*u_pz_*(*k*)]^T^ represents the control value at *k*-*th* control circle. *K_Pb_* is the P coefficient. Δ*s*(*k*) *=* [Δ*x*(*k*), Δ*y*(*k*), Δ*z*(*k*)]^T^ represents position error calculated using Formula (6). *e_p_*(*k*) *=* [Δ*u*_1_(*k*), Δ*v*_1_(*k*), Δ*u*_2_(*k*), Δ*v*_2_(*k*)]^T^ represents position error calculated in image space using the image feature extraction method at *k*-*th* control circle. *e_T_ =* [*e_Tu_*_1_, *e_Tv_*_1_, *e_Tu_*_2_, *e_Tv_*_2_]^T^ represents the set control threshold.
(8)$$\Delta u_{p}(k)=\left\{ \begin{array}{ll} K_{pb}\Delta s(k) & |e_{p}(k)|\geq e_{T}\\ 0 & |e_{p}(k)|\;<e_{T} \end{array}\right.$$

#### 3.2.3. Insertion Depth Control Module Based on LDD

The insertion depth control process is shown in Figure 5. After position fine alignment, the micro-tube position in the X-axis is denoted as *P_xo_*. For insertion depth control, first, based on prior geometric knowledge of the micro-pore, the micro-tube movement (*r_p_*) is controlled in the negative X-axis direction from *P_xo_*, where *r_p_* is the radius of the micro-pore, and the target location is denoted as *P_xl_*. Second, the micro-tube movement is controlled in the Z-axis direction until the micro-tube end contacts the edge of the micro-pore, and the movement length is denoted as *r_h_*. The current position is recorded as *P_ze_*. Third, the micro-tube movement (*r_p_*) is controlled in the X-axis direction to *P_sm_*. Last, micro-tube motion (*l_d_*) is controlled along the Z-axis direction, where *l_d_* is the expected insertion depth.

Whether the micro-tube end contacts the edge of the micro-pore is discriminated in real-time based on an image grayscale variation calculation. If the image grayscale variation is less than the set threshold value (*T*), continue Z-axis direction motion; otherwise, stop Z-axis direction motion and record *P_ze_*. Image grayscale variation is calculated using the background difference method, as shown in Formula (9).
(9)$$R=\sum_{q=1}^{h}\sum_{p=1}^{w}\left(B(p,q)-F(p,q\right))$$
where *B*(*p*, *q*) represent the image gray of the background image, *F*(*p*, *q*) represents the image gray of the foreground image, and (*p*, *q*) represents the image coordinates. *w* and *h* represent the width and height of the detection area.

#### 3.2.4. Glue Mass Control Module

Due to its time–pressure controllability, the dispensing mode is used for dispensing control. However, parameters such as pressure, contact time of the dispensing needle, dispensing needle diameter, etc., all affect glue mass. Thus, the roles of different parameters for glue mass control are analyzed based on simulation.

Simulation model

Based on ANSYS finite element simulation, the volume of fluent model (VOF model) is used to track and calculate the gas-/liquid-free interface.

To improve the accuracy of the simulation and shorten calculation time, the three-dimensional simulation model is simplified into a two-dimensional simulation model, half of which is used as the simulation model, as shown in Figure 6a. Parameters of the simulation model are listed in Table 1.

2Simulation analysis

(1) Influence of pressure on glue mass: Based on the simulation model described above, the angle between the dispensing needle and the micro-tube is set to 45°, glue viscosity is set to 8.5 p, and the dispensing needle diameter is set to 10 μm to study the effect of pressure on the glue mass. The pressure is set to 0 hPa, 60 hPa, 120 hPa, and 180 hPa, respectively. The change of glue mass with time under different pressure conditions is shown in Figure 7a. It can be seen that, for the condition of 0 hPa, the glue mass stabilized over a short period of time. The simulation results for the condition of 0 hPa are shown in Figure 6b. Quantitative glue mass values could be obtained for the condition of 0 hPa.

(2) Influence of dispensing needle diameter on glue mass: For the condition of 0 hPa pressure, the effect of the dispensing needle diameter on glue mass was studied. The diameter of the dispensing needle is set to 5 μm, 10 μm, 15 μm, and 20 μm, respectively. The relationship between diameter and glue mass is shown in Figure 7b. It can be seen that glue mass ultimately able to be steadied under different diameters. The glue mass is increased by the diameter. That is, we can control glue mass quantitatively by controlling the dispensing needle diameter.

Through simulation results listed above, we know that glue mass can be controlled quantitatively for the condition of 0 hPa, and different glue masses can be obtained by changing the diameter of dispensing needle.

## 4. Assembly Experiment

To verify the proposed method, a series of experiments was conducted based on the micro-assembly robot.

The No. 1 and No. 2 microscopic vision system are composed of “1.5× objective + 12× zoom lenses + 2× adapter + 2/3-inch CCD (Baumer TXG50)”. All lenses used were of the Navitar brand. Lens magnification range is from 1.74× to 21×, and lens resolution is from 11.6 µm to 2.2 µm. The No. 3 and No. 4 microscopic vision system are composed of “10× objective + 12× zoom lenses + 2× adapter + 2/3-inch CCD (Baumer TXG50)”. Lens magnification range is from 5.52× to 66.6×, and lens resolution is from 3.55 µm to 1.19 µm.

### 4.1. Posture Alignment Control Experiment

#### 4.1.1. Attitude Fine Alignment

Attitude detection accuracy is the key factor affecting alignment accuracy. Therefore, the attitude detection error analysis is conducted first. The experiment is detailed as follows. As the position accuracy of the micro-manipulator is better than 1 μm and the angle accuracy is better than 0.02 degrees, the movement of the micro-manipulator is considered to be a true value, and vision detection is regarded as the measurement value. In the first verification experiment, only rotate the micro-tube around the X-axis when the other two angles remain unchanged. The relative rotation amount *θ_x_* is used as true the value to compare with the measured value *ϕ_x_*.

Figure 8a shows the contrasting results between *θ_x_* and *ϕ_x_*, and its error is distributed in the interval [−0.207°, 0.009°]. Similarly, the other verified experiment is implemented by comparing *θ_y_* and *ϕ_y_*. As shown in Figure 8b, the error distribution is [−0.292°, 0.047°]. The above results prove that the proposed method can evaluate the attitude components with an accuracy of less than ±0.3°.

Micro-tube attitude is adjusted by the P controller. The control law is shown in Formula (4). The P coefficient is set to 0.5, and the control threshold around the X- and Y-axis are both set to 0.3°. Figure 9a shows the attitude control error in the alignment process. It can be seen that attitude error converged to the threshold range quickly and stably. Figure 10 shows images before and after attitude alignment. The center axis of the micro-tube is extracted to calculate attitude error.

#### 4.1.2. Position Fine Alignment

The position error analysis is similar to the attitude error analysis. By calculating the difference between the real position increment and the detection position increment in Cartesian space, the position detection error is calculated. As shown in Figure 8c, the detection error along the X-axis, Y-axis, and Z-axis are all less than ±4 µm based on 5-group experiment analysis.

Micro-tube position is also adjusted by P controller. Figure 9b shows the position control error in the alignment process. The P coefficient is set to 0.5, and the control thresholds around the X- and Y-axis are both set to 5 µm. Figure 11 shows images before and after alignment, the micro-tube end feature is extracted to calculate position error.

### 4.2. Depth Control Experiment

Figure 8d shows 10 groups of on-line control depths versus off-line measurement results of insertion depth using CT examination. CT examination results are regarded as the true values. Through experimental results analysis, it can be seen that its error is distributed in the interval [−5 µm, 5 µm]. That is, the proposed method can evaluate the insertion depth with an accuracy of ±5 µm.

Through the above experiments, we can see that the attitude control error is less than ±0.3°, the position control error is less than ±5 µm, and the insertion depth control error is less than ±5 µm. The micro-sphere and micro-tube can be aligned with high precision.

### 4.3. Glue Mass Control Experiment

Glue mass control experiments are conducted when dispensing pressure is 0 hPa. The experimental parameters are listed in Table 2.

First, ten repetitions of dispensing are carried out when dispensing pressure is 0 hPa. The results are shown in Table 3; it can be seen for a dispensing needle diameter of 10 μm, the mean value of the glue spot diameter is 62 μm, and deviation of the glue spot diameter is less than 15 μm. The glue spot diameter has good uniformity.

The causes of diameter deviation are analyzed as follows: manufacturing errors of the micro-tube, micro-pore, and dispensing needle, and wall thickness uniformity of the micro-sphere all affect diameter consistency. By improving manufacturing accuracy of the micro-parts, this deviation can be further reduced.

In the case of a cone diameter of 62 μm, cone height is set to half of the diameter, and the glue mass is calculated to be about 31 PL using the geometric calculation method.

Secondly, an experiment of the dispensing needle diameter versus glue spot diameter was carried out. The experimental results are shown in Table 4. It can be seen that the glue spot diameter increases with an increase in the dispensing needle diameter. Different diameters can be selected to get different glue spot diameters based on Table 4.

To summarize: (1) For 0 hPa, using the same-size needle, the glue spot diameter has good consistency. For a 10-μm-diameter needle, the mean value of glue spot diameter is 62 μm, and deviation of glue spot diameter is less than 15 μm. (2) The glue spot diameter increases with increase in the dispensing needle diameter; different dispensing needles can be chosen to meet different glue spot diameter requirements based on Table 4 experimental results.

### 4.4. Assembly Experiment

An entire assembly experiment of CFTA was conducted based on the proposed methods. Figure 12a shows images before alignment. Attitude is finely aligned under the guidance of No. 1 and No. 2, as shown in Figure 12b. Position is finely aligned under the guidance of No. 3 and No. 4, as shown in Figure 12c. The micro-tube is controlled during insertion into the micro-pore, as shown in Figure 12d. The dispensing needle is controlled during movement to the joint, as shown in Figure 12e. Figure 12f shows an image after dispensing. Figure 12g is a CFTA photo image. The CFTA accuracy is measured using CT. The position error in the X, Y, and Z directions is less than 5 μm, and glue spot diameter at the joint is 56 μm. Through volume calculation using the cone calculation formula, the glue mass is about 23 PL when the cone height is half the diameter.

## 5. Conclusions

To achieve high-precision space assembly of a trans-scale micro-device based on micro-vision, CFTA was chosen as the assembly object. Posture alignment control in 5DOF, depth control and bonding control methods of trans-scale micro-parts were studied. First, a micro-assembly robot with four microscopic vision systems was designed for assembly. Second, to achieve high-precision posture alignment in 5DOF, a multi-vision model was established and attitude and position detection methods were designed to align the micro-sphere and micro-tube, for which attitude alignment error can be controlled to less than ±0.3º, and position alignment control error can be controlled to less than ±5 µm. Third, the insertion depth was controlled by a proposed local deformation detection method, and the insertion depth control error was controlled to less than ±5 μm. Fourth, the glue mass at the joint could be controlled quantitatively. Simulation results showed that for 0 hpa, the glue spot diameter was uniform, and using different-size needles, different-size glue spot diameters could be obtained. Experimental results showed that for a dispensing needle with a 10-μm diameter, the mean value of the glue spot diameter was 62 μm, and deviation of diameter was less than 15 μm. A CFTA was assembled based on the proposed approach, the position error of the micro-tube and micro-sphere, as measured by CT, was less than 5 μm, and the glue spot diameter at the joint was 56 μm. Through volume calculation using the cone calculation formula, the glue mass was about 23 PL when the cone height was half the diameter. Furthermore, the methods proposed in the paper will be packaged to apply to other trans-scale micro-device assemblies.

## Figures and Tables

**Figure 1 sensors-23-00450-f001:**
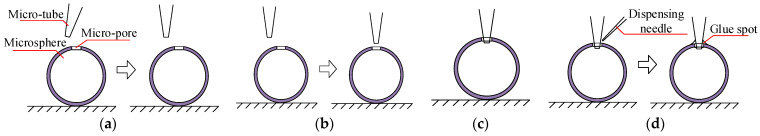
Diagram of assembly process: (**a**) attitude alignment; (**b**) position alignment; (**c**) insertion depth control; (**d**) dispensing process.

**Figure 2 sensors-23-00450-f002:**
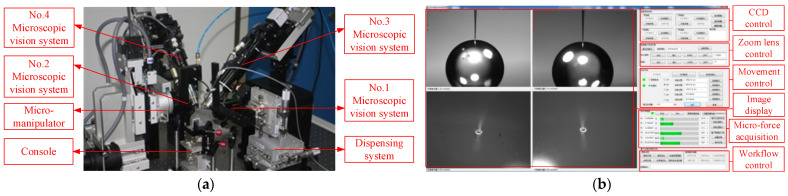
Micro-assembly robot system: (**a**) structure of micro-assembly robot; (**b**) the operation interface of software control system.

**Figure 3 sensors-23-00450-f003:**
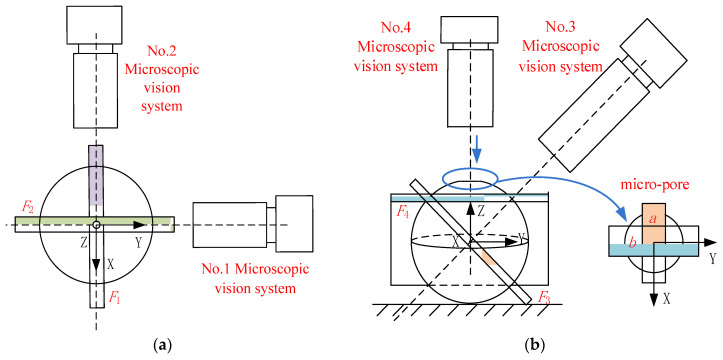
Sketch map of monitoring model: (**a**) horizontal direction; (**b**) 60-degree direction.

**Figure 4 sensors-23-00450-f004:**
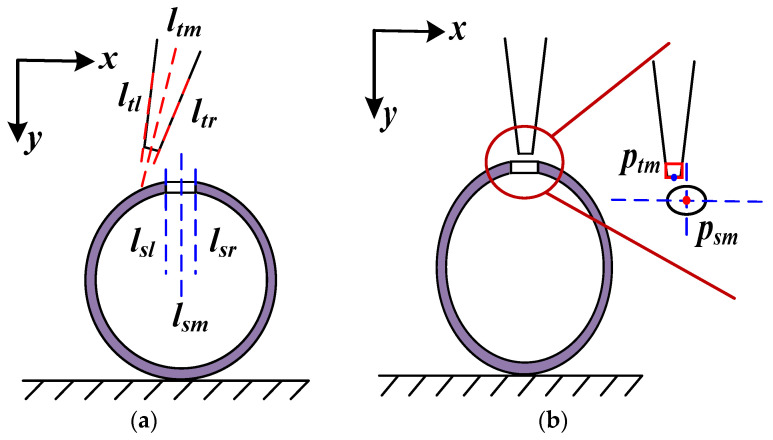
Image feature extraction to calculate attitude and position error: (**a**) feature to calculate attitude error; (**b**) feature to calculate position error.

**Figure 5 sensors-23-00450-f005:**
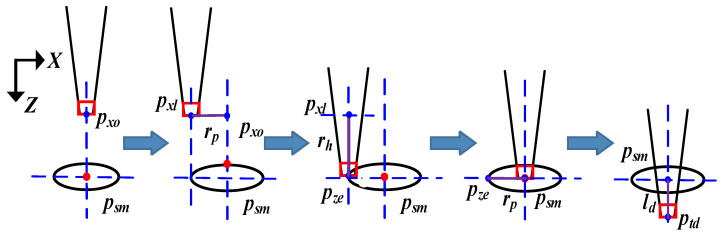
Insertion depth control process.

**Figure 6 sensors-23-00450-f006:**
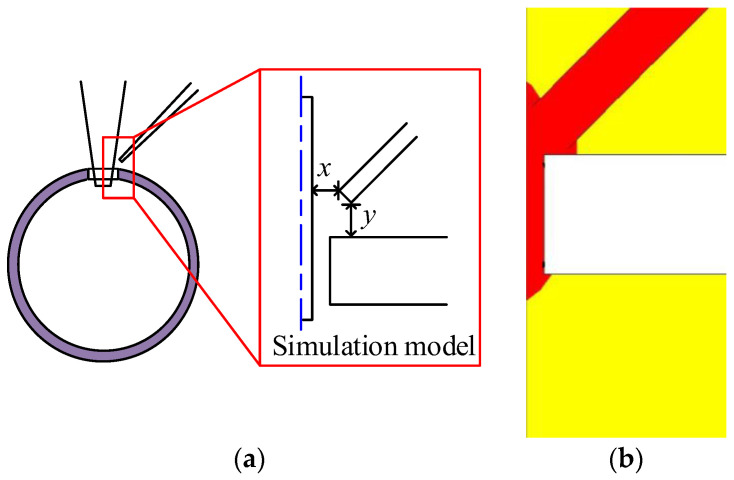
Simulation model and simulation results: (**a**) simulation model; (**b**) simulation results for the condition of 0 hPa.

**Figure 7 sensors-23-00450-f007:**
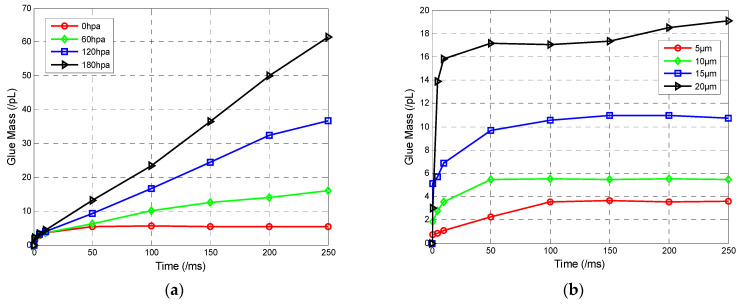
Simulation results: (**a**) pressure versus glue mass, (**b**) diameter versus glue mass.

**Figure 8 sensors-23-00450-f008:**
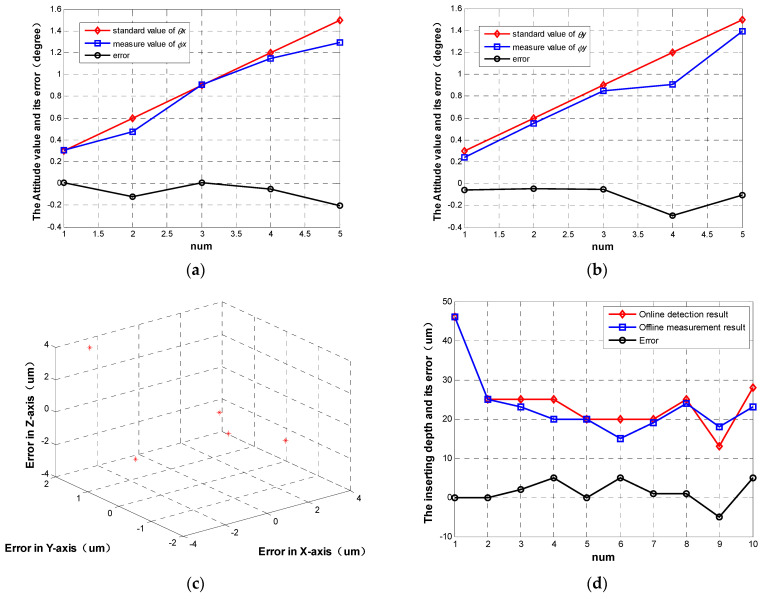
The results of verified experiments for posture detection: (**a**) contrast results between *θ_x_* and *ϕ_x_*; (**b**) contrast results between *θ_y_* and *ϕ_y_*; (**c**) result of verified experiments for position detection; (**d**) depth control error.

**Figure 9 sensors-23-00450-f009:**
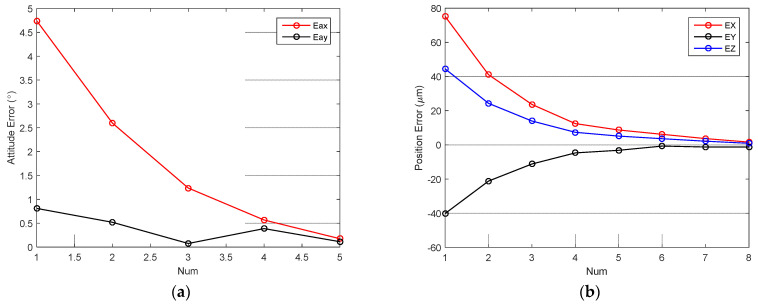
Attitude and position control error: (**a**) attitude error in alignment process; (**b**) position error in alignment process.

**Figure 10 sensors-23-00450-f010:**
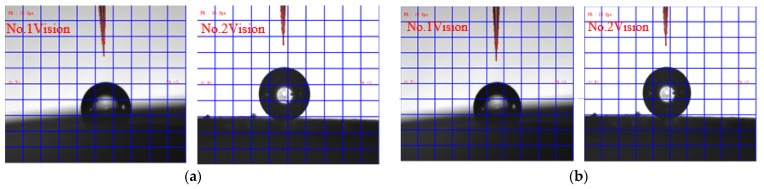
Image of attitude fine alignment: (**a**) images before attitude alignment; (**b**) images after attitude alignment.

**Figure 11 sensors-23-00450-f011:**
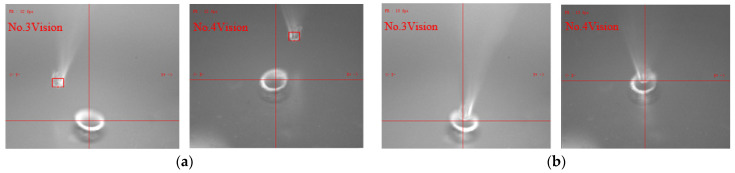
Image of position fine alignment: (**a**) images before alignment; (**b**) images after alignment.

**Figure 12 sensors-23-00450-f012:**
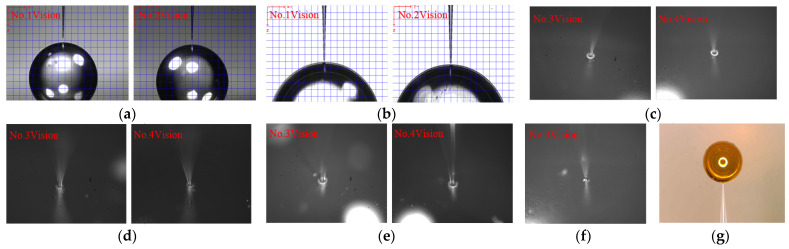
Images in assembly process: (**a**) before alignment; (**b**) after attitude alignment; (**c**) after position alignment; (**d**) after inserting; (**e**) dispensing needle is controlled; (**f**) after dispensing; (**g**) a CFTA photo.

**Table 1 sensors-23-00450-t001:** Parameter of simulation model.

Parameter	Value
Glue density	1220 kg/m^3^
Glue viscosity	8.5 p
Glue surface tension	0.044 N/m
Glue contact angle	33°
Distance x, y	1 μm

**Table 2 sensors-23-00450-t002:** Experimental parameters of dispensing.

Parameter	Value
Glue viscosity	21.38 p
Glue density	1220 kg/m^3^
Glue surface tension	0.044 N/m
Contact angle	33°
Micro-sphere wall thickness	20 μm
Gap between micro-pore and micro-tube	1.5 μm
Distance x	1 μm
Distance y	1 μm
Angle of needle and micro-tube	45°
pressure	0

**Table 3 sensors-23-00450-t003:** Dispensing results of 10-μm needle.

Number	Glue Spot Diameter (µm)
1	66
2	62
3	55
4	66
5	57
6	73
7	60
8	57

**Table 4 sensors-23-00450-t004:** Needle diameter versus glue spot diameter.

Number	Dispensing Needle Diameter (μm)	Glue Spot Diameter (μm)
1	1.8	20.55
2	2.7	22.47
3	4.3	23.04
4	6.8	25.02
5	7.5	41.09
6	8.3	43.7
7	9.6	48.3
8	10.4	51.31
9	11.8	53.23
10	12.6	47.47
11	13.9	49.3
12	14.8	55

## Data Availability

Not applicable.

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
