# Peer review of "Micro-Vision Based High-Precision Space Assembly Approach for Trans-Scale Micro-Device: The CFTA Example"

_sensors, 2023, doi:10.3390/s23010450_

Round 1

Reviewer 1 Report

The article presents about Micro-assembly Robot System for multi-vision system, micro-operating system, dispensing system and software control system.

The error of this device is very small (less than 5µm).

This device has not been evaluated and compared with the technology in depth sensors such as Real Scene D435, MS Kinect V2, ...

what is the resolution of the image obtained from this device

Where is the application of this device, is it in medical endoscopy?

The article also contains a lot of spelling errors

Reviewer 2 Report

Thanks to the authors for your work, which I find interesting, though I have some comments.

- In section 3.2.4, the writing of “the volume fluent model(VOF)” is not accurate enough, and it is better to provide a process diagram of software simulation .

- Abbreviations of glue parameter units should be accurate expression, and Glue viscosity in simulation model should be introduced.

- Language usage should be further revised.
